# Early Diagnosis of COVID-19 Images Using Optimal CNN Hyperparameters

**DOI:** 10.3390/diagnostics13010076

**Published:** 2022-12-27

**Authors:** Mohamed H. Saad, Sherief Hashima, Wessam Sayed, Ehab H. El-Shazly, Ahmed H. Madian, Mostafa M. Fouda

**Affiliations:** 1Radiation Engineering Department, National Center for Radiation Research and Technology (NCRRT), Egyptian Atomic Energy Authority, Cairo 11787, Egypt; 2Engineering Department, Nuclear Research Center (NRC), Egyptian Atomic Energy Authority, Cairo 13759, Egypt; 3Department of Electrical and Computer Engineering, Idaho State University, Pocatello, ID 83209, USA; 4Department of Electrical Engineering, Faculty of Engineering at Shoubra, Benha University, Cairo 11672, Egypt

**Keywords:** COVID-19, CNN hyperparameter optimization, image classification, grid search, learning rate, momentum

## Abstract

Coronavirus disease (COVID-19) is a worldwide epidemic that poses substantial health hazards. However, COVID-19 diagnostic test sensitivity is still restricted due to abnormalities in specimen processing. Meanwhile, optimizing the highly defined number of convolutional neural network (CNN) hyperparameters (hundreds to thousands) is a useful direction to improve its overall performance and overcome its cons. Hence, this paper proposes an optimization strategy for obtaining the optimal learning rate and momentum of a CNN’s hyperparameters using the grid search method to improve the network performance. Therefore, three alternative CNN architectures (GoogleNet, VGG16, and ResNet) were used to optimize hyperparameters utilizing two different COVID-19 radiography data sets (Kaggle (X-ray) and China national center for bio-information (CT)). These architectures were tested with/without optimizing the hyperparameters. The results confirm effective disease classification using the CNN structures with optimized hyperparameters. Experimental findings indicate that the new technique outperformed the previous in terms of accuracy, sensitivity, specificity, recall, F-score, false positive and negative rates, and error rate. At epoch 25, the optimized Resnet obtained high classification accuracy, reaching 98.98% for X-ray images and 98.78% for CT images.

## 1. Introduction

The World Health Organization (WHO) declared coronavirus disease (COVID-19) as a highly contagious illness with a primary reproductive number to be a global pandemic on 11 March 2020. The most recent coronavirus discovery has endangered people’s health and caused significant economic losses worldwide [1]. As of 7 September 2020, the WHO reported that 27,032,617 people had COVID-19 infections due to testing, among which 881,464 died [2]. Recent research shows that the sensitivity ranged from 71% (19 February 2020) to 91% (27 March 2020) and 96% (2022). The specificity of tests for identifying COVID-19 using reverse transcription polymer chain reaction (RT-PCR) analysis is around 95%, according to a recent systematic review. However, the sensitivity of PCR tests for COVID-19 may be in the range of 71–98% (21 April 2020) [3]. The probability of COVID-19 infection is thus not ruled out by one or more negative findings according to the WHO. Therefore, there is an urgent need for further auxiliary tests with somewhat higher sensitivity to COVID-19 [4].

It is known that COVID-19 victims had lung infections, according to autopsies, according to a report from the CHINA–WHO COVID-19 joint investigative panel [5]. Because it uses current medical technology and clinical tests, lung medical imaging is an excellent supplemental diagnostic testing approach for COVID-19. The most frequent medical imaging tests for the lungs are chest computed tomography (CT) and chest X-ray radiography (CXR), both of which are accessible in most hospitals worldwide [5]. Compared to a regular CXR, a CT has better tissue contrast and more gray levels (between 32 and 64) but at a higher cost. This is because a three-dimensional body representation is produced through the computerized processing of CT scans. In addition, recent research directions revealed that using CXR and CT scans increased the diagnostic sensitivity for detecting COVID-19, hence mitigating its spread [6]. However, the interpretation of medical imaging is time-consuming, labor-intensive, and frequently subjective [7]. Moreover, within the present epidemic, frontline expert physicians are overloaded with many diagnoses within a short time, which might impair diagnostic efficiency. Therefore, computerizing COVID-19 detection in hospitals using medical image processing algorithms will accelerate COVID-19 diagnosis with considerable accuracy, reducing the strain on specialists [8,9,10,11].

Deep learning (DL) schemes, particularly convolutional neural networks (CNNs), have made tremendous strides in handling issues like image classification, object identification, gaming, decision making, and natural language processing [12]. However, due to the multi-modal features of clinical data and the DL model’s lack of interpretability, the applications of DL for clinical diagnostics are still limited [13]. Nevertheless, studies have shown that DL approaches work exceptionally well for diagnosing pneumonia using CXR pictures [14], diabetic retinopathy with retinal fundus images [15], and lung cancer with CT images [16,17].

To the best of our knowledge, the DL-aided approach has only been verified on single modal data, with no association study with clinical indications. Traditional machine learning (ML) schemes are more limited and suited to practical computing applications that use features than DL approaches [18].

Any CNN structure contains many design decisions that could greatly affect the performance of the network if it is optimized, such as loss function [19,20] and different network hyperparameters such as:The number of convolutional layers.The number of filters.The filter size.The stride is thus the number of steps the filter takes as it slides across the input image.The kind of padding might be the same or a valid convolution.The batch size.The number of training epochsThe learning rate, momentum, and dropout probability.

Hence, the values of the hyperparameters determine a CNN’s success rate in addressing a specific task. This paper focuses on optimizing the CNN hyperparameters to obtain the highest accuracy for COVID-19 diagnosis. A grid search method will be utilized to optimize the CNN hyperparameters to improve the achieved accuracy. To this end, we created a universal end-to-end DL framework for extracting information from CXR images (X-data) and CT images (CT-data), which may be considered a cross-domain transfer learning model [21]. The key contributions of this paper are summarized as follows:The CNN hyperparameters are optimized using the grid search approach to reduce model losses and achieve the best degree of COVID-19 diagnostic accuracy.The grid search optimization algorithm chooses the optimum CNN hyperparameters from a provided list of parameter possibilities, automating the “trial-and-error” process to obtain the optimized parameters with the greatest diagnosis accuracy.The optimization approach is tested and compared using three different well-known CNN structures (GoogleNet, VGG16, and ResNet50).The optimized CNN architectures were used to classify CT and CXR images for COVID-19 patients to increase the diagnostic sensitivity for identifying COVID-19.Simulation results confirm the efficiency of the CNN architectures with optimized hyperparameters regarding disease categorization. Hence, our envisioned approach outperforms unoptimized CNN methods.Furthermore, our envisioned technique attains the highest accuracy compared to previous techniques using both X-ray and CT images.
This paper is organized as follows. Section 2 highlights the related work. Afterward, Section 3 discusses the proposed work, and Section 4 introduces the leveraged data sets and the reason for utilizing them. Section 5 discusses the experimental results, and finally, the conclusions are highlighted in Section 6.

## 2. Related Work

Several researchers utilized CNNs to detect COVID-19, but the accuracy of their results decreases as the number of CNN layers increases [22]. Therefore, the researcher’s directions turned to possible solutions to enhance CNN performance without altering the design, i.e., using the same layout. The performance and training speed of CNNs are directly influenced by their hyperparameters, such as the size of kernels, number of kernels, length of strides, and pooling size. Additionally, optimizing the hyperparameters’ effects becomes noticeable as the network’s complexity rises. Thus, optimizing the hyperparameters is a useful research direction to enhance the CNN’s performance [23,24].

Recently, deep learning has been widely used for medical applications in general and specifically in medical image-based diagnosis. The improved performance of CNNs in medical image analysis and classification encouraged researchers to use them for a variety of classification tasks related to medical diagnosis. Since the emergence of COVID-19 in December 2019, numerous research studies have been conducted related to using deep learning for COVID-19 diagnosis.

In [25], the authors compared the effectiveness of five pretrained networks (Inception, Xception, VGG16, InceptionResNet50, and MobileNet) to classify X-ray images into three categories (normal, COVID-19, and pneumonia) using a transfer learning strategy. They used two data sets for performance evaluation, including 1427 and 224 X-ray pictures. Their findings demonstrated a DL-based approach’s capability to build an autonomous detecting system employing X-ray images. In [26], the authors developed a DL-based anomaly detection network utilizing X-ray data of 1078 individuals. Despite the achieved specificity of 70.65% and sensitivity of 96%, their model still owns a large number of false positives.

The authors of [27] suggested an approach for categorizing chest CT pictures into two categories (infected and normal) via a CNN. Furthermore, they employed multi-objective differential evolution to set the CNN parameters. In [28], the authors proposed using a DenseNet201 network to classify normal and infected CT images. Their results were promising regards pretrained VGG16 and ResNet50 models. To partition the affected region of the chest, the authors of [29,30] suggested employing U-Net [31] with a residual attention mechanism. A total of 110 axial CT scans from 60 patients were utilized for training their model.

A CoroNet structure based on the Xception architecture was introduced in [32], employing X-ray images to categorize the chest pneumonia diseases such as COVID-19. They reported 87.02% accuracy using limited 1351 images. To categorize X-ray images, the authors of [33] presented a Bayesian DL model with transfer learning. They achieved an accuracy rate of 89% within four different classifications using a data set of 5941 photos. Based on pretrained networks such as VGG16 and InceptionV3, COVIDXNet was first introduced in [34]. A total of 50 X-ray images were collected, and COVID-19 was confirmed in 25 of them. Their approach produced the best result for F1-score at 91%.

In [35], the authors studied the impact of network depth by using a CNN model with extra convolutional layers. Their network was able to achieve a good diagnostic accuracy that reached 99.5% for COVID-19 diagnosis, but they validated their suggestion using a limited-sized data set. In [36], the authors used four variants of optimizers (SGD, Adagrad, Adam, and RMSprop optimizers) in the CNN model to classify the data. The Adam optimizer achieved the best results with an accuracy value of 95.83%. Other network hyperparameters were not optimized. In [37], the authors employed a gradient descent (GD) algorithm to optimize their network parameters. They discussed the basic structure and function and combined back propagation (BP) with the GD method. In [38], the authors optimized four hyperparameters: (1) backbone architecture, (2) the number of inception modules, (3) the number of neurons in the fully connected layers, and (4) the learning rate using the VGG16 CNN Model. CNNs were trained on 2175 computed tomography (CT) images. This method achieved a sensitivity, precision, and accuracy of 97%, 82%, and 88%. In [39], a multi-objective black widow optimization-based CNN (MBWO-CNN) method was proposed. The MBWO method was utilized to tune the CNN hyperparameters, and the extreme learning machine autoencoder (ELM-AE) was leveraged to classify COVID-19. The achieved accuracy result was 97.53%. In [40], five different CNN models were used to extract features. Then, they take the output from each model as input to four different ML algorithms, namely support vector machine (SVM), K-nearest neighbors (k-NN), naive Bayes (NB), and decision tree (DT) for classification. The hyperparameters are optimized by Bayesian optimization to be used for each ML algorithm. This paper’s highest accuracy is 96.29% with the DenseNet201 model and SVM algorithm. However, this method’s processing time for Bayesian optimization is too high.

The lack of a complete automated real-time end-to-end COVID-19 diagnosis system encouraged us to propose our optimized end-to-end trainable framework to achieve the best degree of diagnostic accuracy. This paper focuses on optimizing the CNN hyperparameters employing the grid search approach. The optimized models have been utilized to precisely recognize/diagnose COVID-19 images, including CT and CXR.

## 3. Proposed COVID-19 Detection System

This section explains the suggested optimized model for a COVID-19 detection system. First, different preprocessing procedures are utilized to improve the performance of the CNN model via precise categorization of CXR and CT chest images to find infected individuals. Second, different well-known reference CNN models are to be presented, and their hyperparameters are optimized for better classification accuracy.

Figure 1 shows the flowchart for image classification using optimal CNN hyperparameters. First, the grid search method is leveraged to obtain the optimal learning rate and momentum for more reliable diagnostic performance. Grid search is an exhaustive search approach that targets certain parameter values. The flowchart clarifies five main steps as follows:Load the images and prepare the training and test images.Create an image data augmenter that configures a set of preprocessing options for image augmentation, such as resizing, rotation, and reflection.Resize the training and testing images to the size required by the network.Implement the grid search method to get the optimal learning rate and momentum.Finally, train the model with the optimal learning rate and momentum based on grid search.

### 3.1. Preprocessing Phase

Due to the unavailability of sufficient large data sets for COVID-19 patients, we employed data augmentation techniques in order to generate new data. To this end, we increased the size of our utilized data set. New images are generated by rotating, shearing, zooming, and blurring the original images. The data are then preprocessed in two stages: standardization and normalization. These steps are critical for unifying the data to be fed into the network. Finally, the preprocessed images are passed into the segmentation block to detect picture anomalies and address anomalous aspects and features. In addition to early detection, we allow this block to detect condition severity through further picture processing. Finally, the aberrant pictures are fed into a deep network using the transfer learning approach differentiating COVID-19 from other viral pneumonia diseases. The following sections will provide further information about each block.

#### 3.1.1. Image Augmentation

Data augmentation is used to extract more data from the current data set. In this situation, it makes perturbed duplicates of the existing photos. The principal objective is to strengthen the neural network with different diversities, which results in a network that can differentiate between significant and irrelevant properties in the data set. Several ways may be used to enhance images. When necessary, augmentation methods are effectively used following the quantity and quality of available data. Our approach combines a variety of strategies to support a sizable data set for various situations, as follows:**Gaussian blur**: a Gaussian filter may be used to remove high-frequency elements, resulting in a blurred image version.**Rotation**: a rotation of between 10° and 180° is applied to the picture.**Shear**: using rotation and the imitation factor for the third dimension, picture shearing may be done.

The training set will be expanded using these techniques and utilized to train our optimized CNN models. The testing set will not be expanded. This will demonstrate the architecture’s resilience and how it prevents over-fitting.

#### 3.1.2. Image Processing

An approach to control the complexity and correctness is crucial since the data often originates from diverse sources. Image preprocessing guarantees reduced complexity and improved accuracy when particular data are produced. This approach standardizes the data through several phases to provide the network with a clean data set.

The target of the initial image processing stage is to increase the likelihood of finding the suspicious area. The image’s more delicate features are improved, and noise is eliminated. When medical pictures are distorted by noise, the image’s accuracy is decreased. To reduce this noise, several filters are employed. Background noise is reduced using an anisotropic filter, while salt and pepper noise is reduced using a weighted median filter. Wavelet-based de-noising methods skew wavelet and scaling coefficients.

First, the utilized data set is preprocessed to prepare the used images in a form suitable for the used CNN architectures using the following primary steps:**Image standardization**: it is necessary for CNNs because they deal with images. As a result, the images must first be resized into distinct dimensions and a square form, which is the typical shape used in neural networks (NNs).**Normalization**: To improve the convergence of the training phase, input pixels to any AI system must have a normalized data distribution. To normalize an image, the distribution’s mean value is first subtracted from each pixel, then divided by the result by the standard deviation. Sample X-ray and CT images are shown in Figure 2 before (on the left) and after (on the right) the preprocessing steps.

#### 3.1.3. Segmentation

Threshold methods are used in the segmentation procedure. Segments with similar intensities are produced utilizing the thresholding approach. Images can create borders with the help of thresholding. First, the background noise is checked to see if it is constant or not using an adaptive thresholding approach. For this particular study, the clustering approach is employed. Input data patterns and clustered values are related in clustering procedures. It first groups *n* items into the “*c*” division, where c≤n, according to an attribute. Next, we compare how closely two or more observations match one other. This requires computing the Euclidean distance to achieve similar statements. A measurement of the separation between observations is the Euclidean distance, expressed as follows:(1)$$d_{pq}=\sqrt{{\left(p_{1}-q_{1}\right)}^{2}+{\left(p_{2}-q_{2}\right)}^{2}+{\left(p_{3}-q_{3}\right)}^{2}+\ldots {\left(p_{z}-q_{z}\right)}^{2}}$$

The cluster center is now randomly chosen, indicated by the letter *C*. Next, the fuzzy center is calculated, and finally, the cluster center is updated. A centroid must be determined for each cluster to update the cluster center. It is crucial to compute the centroid by averaging all the *x*- and *y*-axes. It is essential to take the mean of all the points in each cluster to re-compute the new cluster center. Sample X-ray and CT images before (on the left) and after (on the right) the segmentation process are shown in Figure 3.

### 3.2. Reference CNN Models for Transfer Learning

CNNs are very similar to artificial neural networks (ANNs) because they consist of layers and artificial neurons with weights and biases. The main difference between ANNs and CNNs is that CNNs are optimized to work with images. Hence, CNN input is implicitly assumed to be a 2D or 3D image. The name “convolutional” comes from the fact that the first layers of CNN carry out convolutions instead of internal products, as in ANN. CNN is shaped like an ANN by a sequence of layers, but CNN has convolution layers, pooling layers, and fully connected layers [41]. The convolutional layer contains a set of filters (or kernels) that are discrete number grids and is usually square. It performs a convolution operation between these filters and the layer input to create feature maps.

The pooling layer directly comes after the convolution layer. The main objective of this layer is to decrease the dimension (since it combines the neighboring pixels of a particular area of the image in a single value) and, at the same time, highlight the image’s characteristics. Some of the most popular forms of operation are max pooling and mean pooling. The max pooling layer reports the maximal values in each rectangular neighborhood of each input feature point, while the mean pooling layer reports the mean values. The last layer of the architecture of a CNN is a classifier that determines which class the input image belongs to based on the characteristics detected and extracted by the CNN in the convolutional layers and the pooling layers. The fully connected layer is made up of several neurons equal to the number of classes or classifications. To be inserted into fully connected layers, feature maps are flattened into a single 1D vector.

The hyperparameters of the CNN define network structure variables and network parameters. Hyperparameter optimization aims to improve CNN performance in terms of improved accuracy or minimized loss. Numerous studies have shown the impact of hyperparameter tuning on network performance. There are two most widely used techniques for hyperparameter tuning: grid search and random search [42].

Grid search is a systematic process for hyperparameter optimization. Every combination of hyperparameters is specified and evaluated on the model; it is a trial-and-error method. It can be effective at a small number of hyperparameters. As the number of hyperparameters increases, the utilization of computational resources grows exponentially. On the other hand, random search allows researchers to identify hyperparameter values by sampling randomly from the search space. One of the main shortcomings of this technique is that it does not learn from previous tests, and repeated search with the same hyperparameters is possible.

Deep learning has gained remarkable success in computer vision for medical image processing. CNNs, in particular, have been employed for many medical image classification problems due to their successful extraction of image features. There are several types of deep CNNs, including visual geometry group network (VGG-Net); residual network (ResNet); dense convolutional network (DenseNet); Inception, and Xception [43].

**VGG16**: The foundation for the 2014 ImageNet competition entry was the VGG network design (VGG16), which has 16 layers. Five blocks of convolutional layers and three fully linked layers make up VGG16. Convolution uses a filter of size 3 × 3 with stride 1 and padding 1. After each convolution, the ReLU activation function is applied, and the spatial dimensions are decreased by max-pooling with a 2 × 2 filter, stride 2, and no padding.**GoogleNet**: Using inception modules, GoogleNet conducts convolutions with various filter sizes. ImageNet’s 2014 large-scale visual recognition competition (ILSVRC) was known as GoogleNet thanks to its superior performance. It employs four million parameters and has 2 × 2 layers. Layers are deeper with concurrent use of various field widths, and a 6.67% error rate was attained. The ReLu activation function is utilized for all convolutions, including those inside the inception module, and a 1 × 1 filter is applied before the 3×3 and 5×5 convolutions.**ResNet**: ResNet, a 2015 ILSVRC winner, is another name for the residual network. In order to increase the classification accuracy of challenging tasks, very deep models are employed for visual recognition tasks. However, the training procedure becomes more challenging, and accuracy begins to decline as the network depth increases. Skip connections were utilized to add residual learning to solve this issue. In general, layers are placed for training and network learn features at the end of these layers in a convolutional-based deep NN. A residual-based network has a residual link that spans two or more network levels. ResNet accomplishes this goal by linking the *n*th layer to the (n+x)th layer. The 34-layered ResNet solves the problem of accuracy loss in a deeper convolutional network and is simple to train.

#### 3.2.1. Hyperparameters

During the learning process, parameters such as weights or biases are evaluated and improved in a CNN. However, a hyperparameter is a variable whose value is set before training, so it is neither estimated nor corrected. In other words, hyperparameters are the variables that determine the structure of the network (for example, the number of hidden units) and the variables that determine how the network is trained (for example, the learning rate). Therefore, hyperparameters are very important because the performance of the ANN or CNN largely depends on them [44]. The hyperparameters include the number of convolutional layers, the number of filters, and the size of the filters. The stride is the number of steps that the filter takes when sliding through the input image. Additionally, learning rate, momentum, dropout probability, batch size, and the number of training epochs are examples of hyperparameters that could effectively improve the network performance once optimized. [45]. The CNN’s success rate for solving a certain problem depends upon the values of the hyperparameters [46]. Table 1 shows some hyperparameters related to network training.

#### 3.2.2. Grid Search Scheme

Grid search (GS) is the most common technique for learning the hyperparameter configuration space. Grid search may be thought of as a means of brute force testing all possible combinations of the hyperparameters provided to the grid setup. GS works by evaluating the Cartesian product of a user-defined, limited set of values. The high-performing regions cannot be further exploited by GS alone. As a result, the following procedure must be carried out by hand to find the global optimums [48,49,50]:Start with a large search space and phase scale, then limit them based on prior observations of hyperparameter settings that performed well.Repeat multiple times until the best result is obtained.

The main disadvantage of GS is its ineffectiveness in the configuration space of high dimensional hyperparameters since the number of evaluations grows exponentially as the frequency of the hyperparameter rises. The dimensionality curse refers to this exponential growth. Assuming *k* parameters exist and each has n distinct values, the computational complexity grows exponentially at a rate of O(nk). As a result, GS is only an efficient optimization technique when the hyperparameter setting space is constrained.

Grid search is an exhaustive search that is computationally expensive compared to other schemes [51] as it builds a model for every combination of hyperparameters specified and evaluates each model. However, it finds the best way to tune the hyperparameters based on the training set. Since the training phase is carried out offline before the testing phase, it is acceptable to take a longer duration, allowing the network to test all possible combinations and achieve the best performance. Therefore, we used GS to guarantee the highest achieved accuracy during the testing phase rather than reducing complexity and training duration. This guaranteed accuracy will help in preventing the disease’s spread. Due to its ease of implementation and better accuracy with much high reliability and dimensionless, we utilize the GS scheme. Hence, our COVID-19 detection model minimizes manual interactions by doctors and automatically identifies any abnormalities with high accuracy.

#### 3.2.3. Optimized CNN Model

The CNN is regarded as a highly potent technique and has been widely used in many image classification jobs. It has drawn a lot of interest in several fields, including computer vision tasks, image analysis and identification, and object detection. The CNN rapidly pulls characteristics from images, and thanks to its hierarchical structure, it can deal with images in a dynamic way. These layers are rationally ordered in depth, width, and height, respectively. In such a model, a layer’s neurons are only loosely connected to those of the subsequent layer. This number finally decreases to a score with a single vector probability in the output layer. A preprocessed extracted characteristic is employed to improve COVID-19 categorization.

The CNN utilizes several convolutions and pooling layers in the initial stage to extract probable features. The retrieved features’ spatial size is reduced by using the max pooling layer. The overfitting problem is also lessened by the pooling layer. While average pooling considers average values from the feature map produced by the convolutional process, the max pooling layer only considers maximal values. Stride is a term for the pixel spacing that is utilized while pooling images. A fresh “convolved” image is created with features taken from the previous phase after each convolution stage. If an image is denoted by I(x,y) and the filter employed is represented by f(x,y), the transformation that results is
(2)y(i,j)=(I,f)(x,y)=∑∑I(x−u,y−v)f(u,v).

The rectified linear unit (ReLu) is used as an activation function that connects the output and input of each layer via a sophisticated feature mapping. The ReLu is a linear function that either returns zero or accepts positive input values directly.

**Classification**: The flattened layer creates a single, lengthy feature vector and feeds it to the dense, fully connected layer, which transforms the input into a one-dimensional array. Dense layers categorize by using characteristics from an image retrieved from convolutional layers. Typically, the dropout layer decreases the feature map and minimizes overfitting with the aid of the activation function. The final output is predicted using a sigmoid in the final dense layer. The sigmoid function is typically expressed as follows:(3)$$S\left(t\right)=\frac{1}{1+e^{-t}}.$$

## 4. Utilized Data Sets

X-rays and CT are two typical medical imaging modalities used to diagnose and assess the severity of a disease. Each medical imaging technique has its pros and cons. Due to its low cost and considerable availability worldwide, X-ray imaging is the most commonly utilized medical imaging approach for COVID-19 diagnosis. It is processed using elementary techniques, which decreases imaging time and hence the risk of viral propagation. Compared to a CT scan, X-ray imaging is non-invasive and generates minimal radiation exposure. Despite its benefits, X-rays are less sensitive, which may result in a mistaken diagnosis of the illness with early and moderate symptoms. CT scans, on the other hand, are more sensitive and provide precise information about the damaged location, resulting in reliable judgment. Therefore, CT scans are crucial in the diagnosis of lung disorders, and hence are more reliable regarding the early detection of COVID-19. However, its expensive cost, increased radiation dosage, and resource restrictions bound its usage. This paper utilized the publicly accessible pneumonia normal-chest-X-ray posteroanterior (pa) data set [52]. It consists of images gathered from the websites of the Italian Society of Radiology (SIRM), Kaggle, Radiopedia, and Figshare data repository. The data set contains posteroanterior (PA) chest X-ray images divided into three classes (COVID-19, pneumonia, and normal). It consists of 2313 samples for each class, resulting in 6939 samples. Figure 4 displays a sample of that data set.

The China National Center for Bio-information CT chest imaging data set [53] was also employed in our investigations. The images of the data set are classified as coronavirus pneumonia, common pneumonia, and normal. The data set includes 617,775 CT slices from 6752 CT scans for 4154 patients. It consists of CT images for 999 COVID-19 patients, and 1687 normal and 1468 common pneumonia patients. Figure 5 depicts the CT samples of the utilized data sets leveraged in our investigations.

## 5. Results and Discussions

The hyperparameters utilized to optimize the CNN by obtaining the optimal values are learning rate, momentum, and the number of epochs. The grid search method is employed to tune the hyperparameters of ResNet50, VGG16, and GoogleNet to achieve improved performance. Table 2 shows the hyperparameter ranges used to get the optimal values.

The training process was conducted via MATLAB 2019a (MathWorks, Natick, MA, USA), running on a machine with an Intel core i7-4510U CPU @ 2.60 GHz, 8 GB of memory, and a Windows 8.1, 64-bit operating system.

Table 3 lists the optimized hyperparameters obtained using grid search optimization of the three models.

### 5.1. Evaluation Metrics

Different evaluation metrics were employed to reflect the improved performance of our proposed optimization approach. They are listed as follows:**The sensitivity or recall**It is the accuracy of positive examples. It refers to how many examples of the positive classes were labeled correctly. This is shown in Equation (Equation 4), where TP is the true positives, which are the number of instances that are correctly identified, and *FN* is the false negatives, which are the number of positive cases that are classified as negative by mistake [54].
(4)$$\mathrm{Sensitivity}\;\left(\mathrm{Recall}\right)=\;\frac{TP}{TP+FN}$$**Specificity**It refers to the conditional probability of true negatives given a secondary class. It approximates the probability of the negative label being true as in Equation (Equation 5), where TN is the number of true negatives classified as negative and FP is the number of false positives, defined by the negative instances that are classified incorrectly as positive cases. In general, sensitivity and specificity evaluate the positive and negative effectiveness of the algorithm on a single class, respectively [55].
(5)$$\mathrm{Specificity}=\;\frac{TN}{TN+FP}$$**Accuracy**It is used to evaluate classification performance. It calculates the percentage of samples that are classified correctly, as shown in Equation (Equation 6) [55].
(6)$$\mathrm{Accuracy}=\;\frac{TP+TN}{TP+TN+FP+FN}$$**Precision**It is calculated by the number of true positives divided by the number of true positives plus the number of false positives as in Equation (Equation 7). It evaluates the predictive power of the algorithm. Precision is how “precise” the model is out of those predicted positive and how many of them are actually positive [55].
(7)$$\mathrm{Precision}=\frac{TP}{TP+FP}$$**F-score**It focuses on the analysis of positive class. It combines both precision and recall, as shown in Equation (Equation 8). A high value of it indicates that the model performs better on the positive class [55].
(8)$$F_{\mathrm{score}}=2*\frac{\mathrm{Precision}*\mathrm{Recall}}{\mathrm{Precision}+\mathrm{Recall}}$$**False-positive rate and false-negative rate**False-positive rate (FPR) is the portion of negative cases identified improperly as positive instances to the total number of negative instances. FPR is an error in classification in which a test result incorrectly indicates the presence of a condition (such as a disease when the disease is not present), while a false-negative rate (FNR) is the opposite error, where the test result incorrectly indicates the absence of a condition when it is actually present. These are the two kinds of errors in a test, in contrast to the two kinds of correct results (a true positive and a true negative). They are also known in medicine as a false-positive (or false-negative) diagnosis, and in statistical classification as a false-positive (or false-negative) error.
(9)$$\mathrm{FPR}=\frac{FP}{FP+TN}$$
(10)$$\mathrm{FNR}=\frac{FN}{FN+TP}$$The error rate is the performance statistic that informs of incorrect predictions without classifying positive and negative forecasts. It can evaluate by
(11)$$\mathrm{ErrorRate}=\frac{FP+FN}{TP+FP+TN+FN}$$

### 5.2. Results

Table 4 and Table 5 present the performance measures for VGG 16 without and with optimized hyperparameters, respectively. The maximum achieved accuracy of the unoptimized network was 93.52% for COVID-19, 92.89% for viral pneumonia, and 94.86% for normal. On the other hand, optimizing the same network using our proposed hyperparameter optimization outperformed the unoptimized network and achieved an improved accuracy of 97.69% for COVID-19, 97.82% for viral pneumonia, and 98.91% for the normal class.

Table 6 and Table 7 present the performance measures for GoogleNet without and with optimized hyperparameters, respectively. The maximum achieved accuracy of the unoptimized network was 96.52% for COVID-19, 96.39% for viral pneumonia, and 96.86% for normal. On the other hand, optimizing the same network using our proposed ideas outperformed the unoptimized network and achieved an improved accuracy of 98.49% for COVID-19, 98.18% for viral pneumonia, and 98.93% for the normal class.

Table 8 and Table 9 present the performance measures for ResNet-50 without and with optimized hyperparameters, respectively. The maximum achieved accuracy of the unoptimized network was 96.84% for COVID-19, 96.87% for viral pneumonia, and 97.58% for normal. On the other hand, optimizing the same network using our proposed ideas outperformed the unoptimized network and achieved an improved accuracy of 98.98% for COVID-19, 98.75% for viral pneumonia, and 99.84% for normal. Optimizing the hyperparameters using our proposed has a positive impact on the network performance in terms of training time and accuracy.

Moreover, the optimized ResNet model for X-ray images was fine-tuned using the CT chest imaging data set via transfer learning, and the obtained results are listed in Table 9. It is clear that the performance was improved in terms of achieved accuracy and time required for training as the fine-tuned model was capable of achieving excellent accuracy (99.79%) for COVID-19, 98.53% for viral pneumonia, and 99.79% for the normal class.

Table 10 present the performance measures for ResNet-50 with optimized hyperparameters for the CT data set. The maximum achieved accuracy is 98.78% for COVID-19, 98.53% for viral pneumonia, and 99.79% for normal.

The results in Table 4, Table 5, Table 6, Table 7, Table 8, Table 9 and Table 10 show that our proposed optimization strategy for obtaining the optimal learning rate and momentum of the CNN’s hyperparameters using the grid search method succeeded in improving the network performance for both X-ray or CT data sets [52].

Furthermore, Table 11 and Table 12 show the performance comparison of our proposed technique with previous related work in terms of detection accuracy for the X-ray and CT data sets, respectively. As shown in the two tables, our envisioned method achieves the highest accuracy compared to others confirming its efficiency and accurate results.

Finally, Table 13 provides the computational time for optimizing the hyperparameters for three different models used in the study, it was added to the revised manuscript.

## 6. Conclusions

The early detection of COVID-19 is a vital step to preventing its spread. CNN-aided solutions can help in the extraction of essential features from X-ray and CT images to assist in the early diagnosis of COVID-19. In this paper, we present an efficient hyperparameter optimization procedure that can efficiently improve the classification accuracy of well-known CNN models. We employed the grid search method to get the optimal learning rate and momentum with different epoch numbers for different retrained networks to be used for more reliable diagnostic performance. Compared to unoptimized models, our optimization approach has a great impact on the performance of the used models. Simulation results confirm the efficiency of the proposed techniques in terms of achieved accuracy, sensitivity, specificity, precision, recall, F-score, FPR, and FNR, leading to precise disease detection. In terms of achieved accuracy, our proposed optimization approach outperforms all recent state-of-the-art techniques, achieving classification accuracies of 98.98% and 98.78% for the X-ray and CT data sets, respectively.

## Figures and Tables

**Figure 1 diagnostics-13-00076-f001:**
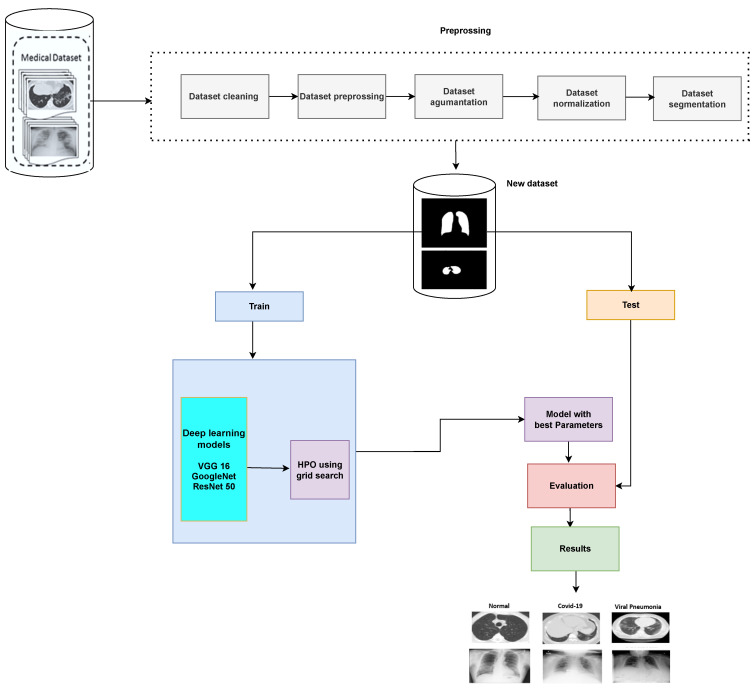
Image classification using optimal CNN hyperparameters.

**Figure 2 diagnostics-13-00076-f002:**
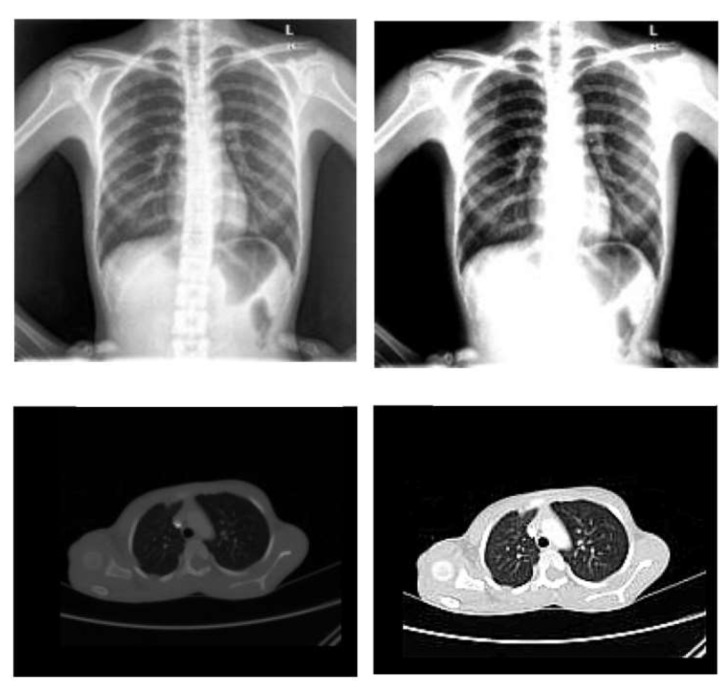
Sample X-ray and CT images before and after the preprocessing steps.

**Figure 3 diagnostics-13-00076-f003:**
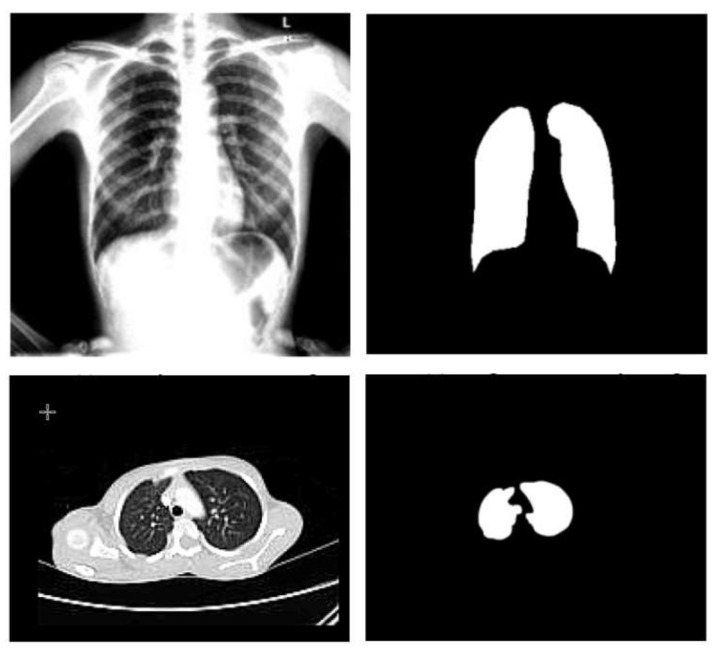
Sample X-ray and CT images before and after the segmentation step.

**Figure 4 diagnostics-13-00076-f004:**
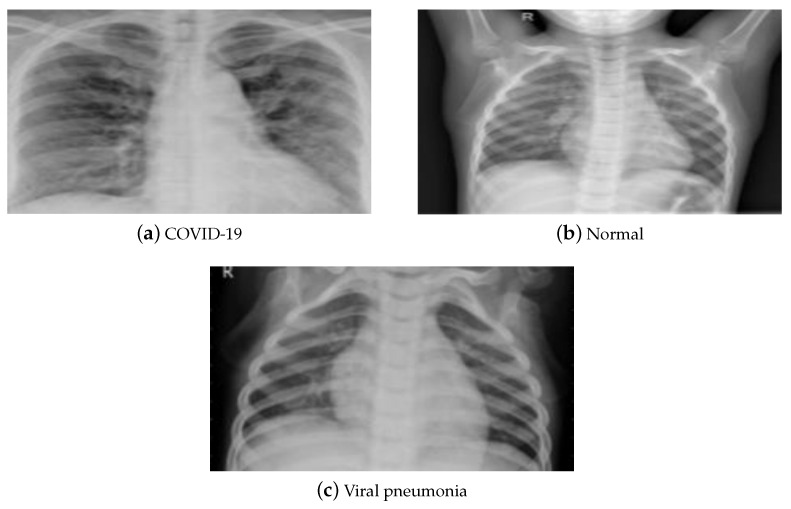
(**a**–**c**) X-ray data set samples [52].

**Figure 5 diagnostics-13-00076-f005:**
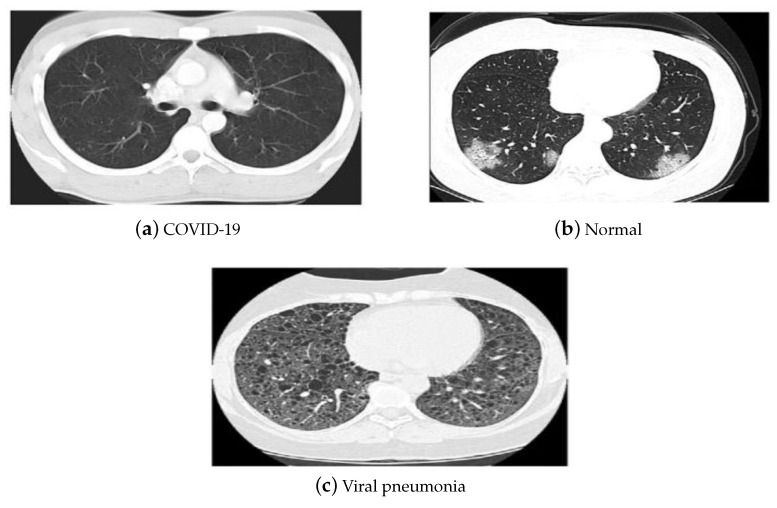
(**a**–**c**) CT data set samples [53].

**Table 1 diagnostics-13-00076-t001:** Hyperparameters related to the training of a neural network [47].

Hyperparameters	Description
Learning Rate	The learning rate defines how quickly a network updates its parameters. For the classification problem, it is important to choose the optimal learning rate to minimize the loss function. A low learning rate slows down the learning process but converges smoothly. A larger learning rate speeds up the learning but may not converge.
Momentum	Momentum helps to know the direction of the next step with the knowledge of the previous steps. It helps to prevent oscillations.
Number of Epochs	The number of epochs is the number of times the whole training data are introduced to the network. It is important to determine an ideal epoch number to prevent overfitting.
MiniBatch Size	The larger minibatch size causes running of the model for a long period of time with constant weights that causes overall performance loses and increases the memory requirements. Carrying out the experiments with small minibatch sizes can be more beneficial.

**Table 2 diagnostics-13-00076-t002:** Utilized hyperparameters range to obtain optimal CNN.

Hyperparameters	Range
Learning Rate	[0.09, 0.07, 0.05, 0.03, 0.01]
Momentum	[0.9, 0.7, 0.5, 0.3, 0.1]
Number of Epochs	10 to 100 with step 5

**Table 3 diagnostics-13-00076-t003:** Optimized hyperparameters for the three models.

Hyperparameters	Resnet	Google Net	VGG16
Learning Rate	0.01	0.01	0.02
Momentum	0.1	0.1	0.3
Number of Epochs	25	30	45
Activation Function	Relu	Relu	Relu
Classifier	Softmax	Softmax	Softmax
Loss Function	Cross entropy	Cross entropy	Cross entropy

**Table 4 diagnostics-13-00076-t004:** Performance measures for VGG 16 without optimization for X-ray images.

Category	Accuracy (%)	Specificity (%)	Precision (%)	Recall (%)	F1-Score (%)	FPR	FNR	Error Rate
Normal	94.86	94.91	93.52	94.24	93.88	0.0509	0.0576	0.0514
COVID-19	93.52	93.36	92.67	94.38	93.52	0.0664	0.0562	0.0648
Viral Pneumonia	92.89	94.82	94.83	90.14	92.43	0.0518	0.0986	0.0711

**Table 5 diagnostics-13-00076-t005:** Performance measures for VGG 16 after optimization for X-ray images.

Category	Accuracy (%)	Specificity (%)	Precision (%)	Recall (%)	F1-Score (%)	FPR	FNR	Error Rate
Normal	98.91	98.65	97.67	97.59	97.63	0.0253	0.0648	0.0218
COVID-19	97.69	96.25	96.57	96.78	96.67	0.0135	0.0241	0.0109
Viral Pneumonia	97.82	97.47	98.18	93.52	95.79	0.0375	0.0322	0.0231

**Table 6 diagnostics-13-00076-t006:** Performance measures for GoogleNet without optimization for X-ray images.

Category	Accuracy (%)	Specificity (%)	Precision (%)	Recall (%)	F1-Score (%)	FPR	FNR	Error Rate
Normal	96.86	96.91	95.52	96.24	95.88	0.0309	0.0376	0.0314
COVID-19	96.52	96.51	96.67	96.38	96.52	0.0349	0.0362	0.0348
Viral Pneumonia	96.39	96.82	96.83	96.14	96.48	0.0318	0.0386	0.0361

**Table 7 diagnostics-13-00076-t007:** Performance measures for GoogleNet after optimization for X-ray images.

Category	Accuracy (%)	Specificity (%)	Precision (%)	Recall (%)	F1-Score (%)	FPR	FNR	Error Rate
Normal	98.93	98.88	98.93	98.54	98.73	0.0112	0.0146	0.0107
COVID-19	98.49	98.29	98.78	98.88	98.83	0.0171	0.0112	0.0151
Viral Pneumonia	98.18	98.28	98.58	97.12	97.84	0.0172	0.0288	0.0182

**Table 8 diagnostics-13-00076-t008:** Performance measures for ResNet-50 without optimization for X-ray images.

Category	Accuracy (%)	Specificity (%)	Precision (%)	Recall (%)	F1-Score (%)	FPR	FNR	Error Rate
Normal	97.58	97.56	97.52	97.47	97.49	0.0265	0.0365	0.0313
COVID-19	96.84	96.92	96.95	96.87	96.91	0.0244	0.0253	0.0242
Viral Pneumonia	96.87	97.35	97.27	96.35	96.81	0.0308	0.0313	0.0316

**Table 9 diagnostics-13-00076-t009:** Performance measures for ResNet-50 after optimization for X-ray images.

Category	Accuracy (%)	Specificity (%)	Precision (%)	Recall (%)	F1-Score (%)	FPR	FNR	Error Rate
Normal	99.84	99.79	99.48	99.57	99.52	0.0021	0.0043	0.0016
COVID-19	98.98	98.88	98.78	98.88	98.83	0.0112	0.0112	0.0102
Viral Pneumonia	98.75	98.95	98.58	98.49	98.53	0.0105	0.0151	0.0125

**Table 10 diagnostics-13-00076-t010:** Performance measures for optimized ResNet-50 for CT images.

Category	Accuracy (%)	Specificity (%)	Precision (%)	Recall (%)	F1-Score (%)	FPR	FNR	Error Rate
Normal	99.79	99.65	99.25	99.36	99.30	0.0118	0.0162	0.0147
COVID-19	98.78	98.75	98.63	98.59	98.61	0.0035	0.0064	0.0021
Viral Pneumonia	98.53	98.82	98.58	98.38	98.48	0.0125	0.0141	0.0122

**Table 11 diagnostics-13-00076-t011:** X-ray data set performance comparison.

Reference	Technique	Accuracy (%)
[56]	Resnet50	92.74
[57]	CNN	93.37
[25]	Transfer Learning	95.23
[58]	CNN	95.92
This paper	Optimal CNN Hyperparameters	98.98

**Table 12 diagnostics-13-00076-t012:** CT data set performance comparison.

Reference	Technique	Accuracy (%)
[59]	Random forest	88.6
[60]	3D-CNN	90.7
[61]	Statistical Analysis	89.9
[62]	CNN Future Fusion	94.8
This paper	Optimal CNN Hyperparameters	98.78

**Table 13 diagnostics-13-00076-t013:** Computational time for optimizing the hyperparameters for different models.

Data Type	Model with Optimal Parameters	Time (s)
X-ray	VGG16	26.451
X-ray	Google Net	22.372
X-ray	ResNet	20.003
CT	ResNet	21.145

## Data Availability

Data is available upon a request to the authors.

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
