# Peer review of "Early Diagnosis of COVID-19 Images Using Optimal CNN Hyperparameters"

_diagnostics, 2022, doi:10.3390/diagnostics13010076_

Round 1

Author Response

We appreciate you and the reviewers for your precious time in reviewing our paper entitled (Early Diagnosis of COVID-19 Images Using Optimal CNN Hyperparameters) and providing valuable comments. It was your valuable and insightful comments that led to possible improvements in the current version. The authors have carefully considered the comments and tried our best to address every one of them. We hope the manuscript, after careful revisions, meet your high standards. The authors welcome further constructive comments, if any.

Below we provide the point-by-point responses. All modifications in the manuscript have been highlighted in red.

Reviewer 2 Report

1. The specificity of tests for identifying COVID-19 using RT-PCR analysis is around 95%, according to a recent systematic review. However, the sensitivity of the polymers chain reaction (PCR) test for COVID-19 may be in the range of 71-98% (Apr 21, 2020). The probability of COVID-19 infection is thus not ruled out by one or more negative findings, according to the WHO. Therefore, there is an urgent need for further auxiliary tests with somewhat higher sensitivity to COVID-19.

2. Computerizing COVID-19 detection in hospitals using medical image processing algorithms will accelerate COVID-19 diagnosis with considerable accuracy, reducing the strain on specialists.

3. Studies have shown that DL approaches work exceptionally well for diagnosing pneumonia using CXR pictures, diabetic retinopathy with retinal fundus images, and lung cancer with CT images.

4. The CNN hyper-parameters are optimized using the grid search approach to reduce model losses and achieve the best degree of COVID-19 diagnostic accuracy.

5. Simulation results 434 confirm the efficiency and accuracy of the proposed techniques, leading to precise disease detection.

6. The following papers should be cited:

Aslan MF, Sabanci K, Durdu A, Unlersen MF. COVID-19 diagnosis using state-of-the-art CNN architecture features and Bayesian Optimization. Comput Biol Med. 2022 Mar;142:105244. doi: 10.1016/j.compbiomed.2022.105244. Epub 2022 Jan 20. PMID: 35077936; PMCID: PMC8770389.

Hyperparameter Optimization for COVID-19 Pneumonia Diagnosis Based on Chest CT
  • March 2021
  • Sensors 21(6):2174
  • DOI: 
  • 10.3390/s21062174

  • License
  • CC BY 4.0
  • Paulo Cezar Lacerda Neto
  • Bruno Barros
  • Célio Vinicius N. Albuquerque
  • Aura Conci
An Experimental Approach to Diagnose Covid-19 Using Optimized CNN
  • January 2022
  • Intelligent Automation and Soft Computing 34(2):1065-1080
  • DOI: 
  • 10.32604/iasc.2022.024172

  • Lab: 
  • Anjani Kumar Singha's Lab
  • Anjani Kumar Singha
  • Nitish Pathak
  • Neelam Sharma
  • Show all 8 authors
  • Guthikonda Nagalaxmi

Performance Analysis of Hyperparameters of Convolutional Neural Networks for COVID-19 X-ray Image Classification

Aijaz Ahmad Reshi, Furqan Rustam, Arif Mehmood, Abdulaziz Alhossan, Ziyad Alrabiah, Ajaz Ahmad, Hessa Alsuwailem, Gyu Sang Choi, "An Efficient CNN Model for COVID-19 Disease Detection Based on X-Ray Image Classification", Complexity, vol. 2021, Article ID 6621607, 12 pages, 2021. https://doi.org/10.1155/2021/6621607

6. I am not an expert in medical science but the research method used in this paper followed the standard practices. How is your work different from the cited articles in the list that I generated and recorded above. I am asking about research gap and how you have addressed it.

Author Response

We appreciate you and the reviewers for your precious time in reviewing our paper entitled (Early Diagnosis of COVID-19 Images Using Optimal CNN Hyperparameters) and providing valuable comments. It was your valuable and insightful comments that led to possible improvements in the current version. The authors have carefully considered the comments and tried our best to address every one of them. We hope the manuscript after careful revisions meet your high standards. The authors welcome further constructive comments if any.

Below we provide the point-by-point responses. All modifications in the manuscript have been highlighted in red.

Round 2

Reviewer 1 Report

Accepted